# Efficacy of Souroubea-Platanus Dietary Supplement Containing Triterpenes in Beagle Dogs Using a Thunderstorm Noise-Induced Model of Fear and Anxiety

**DOI:** 10.3390/molecules26072049

**Published:** 2021-04-03

**Authors:** Aleksandar Masic, Gary Landsberg, Bill Milgram, Zul Merali, Tony Durst, Pablo Sanchez Vindas, Mario Garcia, John Baker, Rui Liu, John Arnason

**Affiliations:** 1Department of Microbiology, Educons University, Vojvode Putnika 87, Sremska Kamenica 21208, Serbia; 2CanCog Technologies Inc., P.O. Box 248, Toronto, ON N1M 2W8, Canada; garyl@cancog.com (G.L.); billm@cancog.com (B.M.); 3Brain and Mind Institute, the Aga Khan University, 3 Park Place, Nairobi 00100, Kenya; zul.merali@aku.edu; 4Department of Chemistry and Biomolecular Sciences, University of Ottawa, Ottawa, ON K1N 6N5, Canada; tdurst@uottawa.ca; 5Herbario Juvenal Valerio Rodriguez, Universidad Nacional, Heredia 3000, Costa Rica; pesanche12@gmail.com (P.S.V.); mario78@gmail.com (M.G.); 6Stonehedge Bioresources Inc., Stirling, ON K8N 5J2, Canada; cbdbakerinc@gmail.com; 7Department of Biology, University of Ottawa, Ottawa, ON K1N 6N5, Canada; rliu@uottawa.ca

**Keywords:** triterpenes, anxiety, Souroubea, Platanus, dietary supplement, noise aversion, canines

## Abstract

A novel botanical dietary supplement, formulated as a chewable tablet containing a defined mixture of *Souroubea* spp. vine and *Platanus* spp. Bark, was tested as a canine anxiolytic for thunderstorm noise-induced stress (noise aversion). The tablet contained five highly stable triterpenes and delivered 10 mg of the active ingredient betulinic acid (BA) for an intended 1 mg/kg dose in a 10 kg dog. BA in tablets was stable for 30 months in storage at 23 °C. Efficacy of the tablets in reducing anxiety in dogs was assessed in a blinded, placebo-controlled study by recording changes in blood cortisol levels and measures of behavioral activity in response to recorded intermittent thunder. Sixty beagles were assigned into groups receiving: placebo, 0.5×, 1×, 2×, and 4× dose, or the positive control (diazepam), for five days. Reduction in anxiety measures was partially dose-dependent and the 1× dose was effective in reducing inactivity time (*p* = 0.0111) or increased activity time (*p* = 0.0299) compared with placebo, indicating a decrease in anxiety response. Cortisol measures also showed a dose-dependent reduction in cortisol in dogs treated with the test tablet.

## 1. Introduction

In our research program on botanicals with anxiety-reducing properties, we discovered that extracts of neotropical vines, *Souroubea sympetala* V.A. Richt and *Souroubea gilgii* Gilg. (Marcgraviaceae), had potent anxiety-reducing properties in several rat models of anxiety, including elevated plus-maze, fear-potentiated startle and social interaction [1]. Through bioassay-guided fractionation [2], the active fraction was found to contain five triterpenes (Figure 1). Betulinic acid was found to be the most active component, but a combination of betulinic acid and amyrins was significantly more effective in reducing behaviors associated with anxiety than betulinic acid alone. Other phytochemicals found in whole extracts included hydroxyursolic acid, taraxenyl trans-4-hydroxy-cinnamate, and 2-α-hydroxyursolic acid, isolated as its methyl ester, 2-α-hydroxymaslinic acid [2]. Further investigation showed that the treatment was able to reduce levels of the stress-induced hormones cortisol and corticosterone to normal levels in restrained rodents, fish, and suckling pigs [3,4]. Using antagonists in vivo, the plant extracts and active principles were shown to act, at least in part, at the GABA_A_-benzodiazepine receptor. However, unlike valium, there was no effect on muscle tone, or withdrawal, after prolonged 28 day administration in rodents or suckling pigs [5].

Efforts were then made to develop a practical botanical dietary supplement for animal health by combining dried, ground *S. sympetala* vine with a second botanical, *Platanus occidentalis* L. (Platanaceae) (sycamore) bark, containing mainly betulinic acid (Figure 2). The addition of sycamore improved the anxiolytic efficacy and reduced cost [4]. Various ratios of the two plants were tested in rodent anxiety models and the optimal ratio was determined to be 45:55 w:w. Platanus bark is a botanical used for other applications in traditional human medicines. After the safety of the botanical combination was confirmed in a feeding trial with rodents, the individual plant species and the combined plant species were formulated for dogs as an oral tablet that contained only raw plant material, a beef flavoring extract, and a binding agent. A validated procedure for quality control of the five triterpene ingredients was developed by HPLC-MS [6]. The tablet was formulated to deliver 10 mg of betulinic acid in one tablet for a 10 kg dog, or 20 mg in two tablets for a 20 kg dog, i.e., at a target dose of 1 mg/kg. In a pilot safety study with single dogs, the tablets were orally administered daily for 28 days at 10× intended daily dose and dogs were carefully monitored, throughout the day, for potential signs of toxicity [7]. They were found to produce no significant adverse behavioral effects or significant changes in blood and urine parameters, or clinical observations (weight, behavior, etc.), compared with the control group. This result was confirmed in a larger safety trial (*n* = 6/treatment), wherein beagles were given the Souroubea Platanus tablet at 0×, 0.5×, 1×, and 5× the intended dose, administered orally [8]. Efforts to produce plants sustainably by agriculture were successful [9].

Although a number of anxiety conditions occur in dogs, we chose to target a major problem for owners—noise aversion. Thunderstorms, fireworks, and other loud noises can cause great stress and anxiety in otherwise normal animals. This temporary condition is a suitable application for an effective dietary supplement treatment. In a survey of dog owners, Reiner [10] reported that they had some success with behavioral techniques, but the existing dietary supplements available at the time provided relatively low levels (27–35%) of effective treatments.

Due to the significant anxiolytic effects of Souroubea Platanus seen in other animals, we tested its effectiveness in a well-established thunderstorm model [11,12] with a group of beagle dogs, representative of companion dogs of mixed ages and temperaments. Dogs were monitored for 9 min in a specially constructed room for behavioral measures before (3 min), during (3 min), and after (3 min) a recording of thunderstorm sounds was played. Serum cortisol levels were measured as a stress biomarker prior to entering the room and immediately following test completion. Activity measures (distance travelled, activity, and inactivity) were collected and analyzed using Ethnovision XT (Noldus, Wageningen, The Netherlands) system activity software.

Efficacy of the Souroubea-Platanus tablets in reducing anxiety in dogs was assessed in a blinded, placebo-controlled, noise-induced anxiety model. Sixty beagle dogs were assigned into six experimental groups, each receiving: placebo, 0.5×, 1×, 2×, or 4× the recommended dose, or the positive control (diazepam). Treatments were administered once daily for five days (the 4 days prior to and the day of testing) prior to the thunderstorm anxiety test. Dogs were tested for changes in blood cortisol levels and objective measures of behavioral activity in response to the recorded intermittent thunder.

## 2. Results

### 2.1. Tablet Stability

Tablets analyzed (Table 1) on 14 March 2016 and 14 October 2018 (36 months later) had no significant change in mean weight and concentration of the active principles betulinic acid and α-amyrin following long-term storage at 23 °C.

### 2.2. Activity/Inactivity Time Study during Simulated Thunderstorm with Five Consecutive Daily Doses

During the treatment phase with 5 consecutive days of either placebo, diazepam, or 0.5×, 1×, 2×, or 4× doses of Souroubea-Platanus canine tablets, dogs receiving the positive control diazepam had significantly lower inactivity time scores (for behaviors such as freezing) than placebo-treated dogs did, as expected, indicating lower anxiety than in the placebo group (Table 2). For dogs in the groups receiving 0×, 0.5×, 1×, and 2× doses, the mean (standard error) change in inactivity score was dose-dependent by linear regression analysis (change in activity = 23.8 (±2.7) × dose, *n* = 4, r^2^ = 0.79). In addition, dogs receiving Souroubea-Platanus tablets at doses of 1× and 2× had significantly less inactivity time compared with dogs receiving placebo (*p* = 0.0111 and *p* = 0.0084, respectively). Dogs in the 4× group did not follow this trend, exhibiting greater total inactivity than dogs receiving 1× or 2× activity, although trending below the placebo level.

A second measure of anxiety, the activity score, focused on normal behaviors. As expected, in the diazepam-treated group the activity score was significantly higher than that of the placebo-treated group. Although there was no clear dose-response relationship, dogs receiving 1× and 4× doses had significantly greater activity time compared with dogs receiving placebo (*p* = 0.0299 and *p* = 0.0199, respectively) and marginal significance was seen in dogs receiving 2× dose (*p* = 0.0942). At the lowest dose (0.5×), there was no significant change.

### 2.3. Cortisol Results

Groups of beagles were identified as cortisol responders when the serum cortisol was elevated by >13% following exposure to thunder sounds, compared with pretest baseline levels (i.e., before any drugs were administered (Figure 3a)). After the drug was administered in these dogs, the mean prethunder cortisol levels were not different between the control group, diazepam-treated dogs, or those treated with different doses of Souroubea-Platanus tablets. After exposure to simulated thunder, cortisol levels in the control dogs doubled from 46 to 90 nmol/L, a clear indication of a stress response. Diazepam lowered cortisol to prethunder levels, as expected. The Souroubea-Platanus tablet had a dose–response effect on reductions to cortisol, with the highest dose demonstrating reductions similar to diazepam. Cortisol levels did not fall below prethunder control levels in any treatment group.

## 3. Discussion

The results showed that the main anxiolytic triterpenoid compounds, betulinic acid and α-amyrin, were stable in the tablet for 30 months—an important consideration in an efficacy trial and for future practical applications. Stability in botanicals is a concern, as in our experience some active principles, such as alkamides in echinacea, decline significantly in concentration in a period of 6–12 months.

The results showed that orally administered Souroubea-Platanus tablets reduced both behavioral and biochemical outcome measures which assessed the effect of the sound of thunder on anxiety and stress in dogs. This study compared the effectiveness of Souroubea-Platanus tablets at four dose levels with that of diazepam and a placebo control. The botanical treatment was significantly effective in all measures at the intended 1× dose, reducing inactivity and increasing activity measures in all dogs and cortisol in the responding dogs, compared with placebo-treated dogs. The botanical treatment showed a significant dose–response effect on inactivity measures and an obvious dose-dependent trend in cortisol results in responding dogs.

Cortisol is an objective biochemical measure of the stress response and does not require observer assessments. There was a clear dose–response effect in lowering post-thunder cortisol compared with the elevated levels seen in the control. The botanical treatments were somewhat less effective than the diazepam treatment for cortisol lowering, but this is to be expected with a botanical dietary supplement and may be desirable for treatment of otherwise healthy dogs, as the treatment did not cause the well-known, undesirable effects of valium (withdrawal symptoms and motility impairment) in 28 day treatments of rodents.

Collectively, these findings provide support for the use of Souroubea-Platanus tablets in reducing noise-induced anxiety in dogs, by demonstrating their effectiveness in reduced cortisol release, increased activity measures in response to thunder and reduced inactivity measures. In terms of future work, Souroubea-Platanus tablets may have effectiveness in reducing conditioned anxiety in dogs, as has been demonstrated in rat models [13]. Further studies on botanicals with higher amyrin content may improve potency.

Overall, this study suggests that *Souroubea* spp. and *Platanus* spp. exert anxiolytic effects in dogs by inhibiting cortisol release and helping to maintain normal behavior. The results also suggest that the lowest effective dose in dogs is the currently recommended dosage (i.e., 1× dose). The results support their use as an effective and natural anxiolytic for dogs during periods of exposure to loud noises, such as thunder.

Until recently, the development of evidence-based botanicals for the animal healthcare market has lagged behind the human healthcare market. With this study on Souroubea Platanus and recent publications, such as Mastinu et al. [14] showing the protective effect of *Gynostemma pentaphyllum* against lipopolysaccharide-induced inflammation and motor alteration, the botanical options for animal healthcare are increasing.

## 4. Materials and Methods

### 4.1. Drugs

Diazepam (Valium, Lot No. S12C02) was used as a positive control article and was administered at a dose of 0.5 mg/kg.

### 4.2. Tablet Preparation and Analysis

The growth and collection of plant materials and their formulation into a 3.5 g chewable tablet has been described in detail elsewhere [12]. The *Souroubea sympetala* was identified by taxonomist P. Sanchez (Voucher# OTT19994) and plant material was sustainably harvested from agricultural production of three-year-old vines in Costa Rica. Wild collection of *Platanus occidentalis* (voucher #19608) shedding bark was undertaken in Niagara and Essex counties, Ontario, without injury to trees [9]. The tablets were formulated in Guelph, Ontario, with 55:45 w:w dry powdered *Souroubea sympetala* vine: *Platanus occidentalis* bark, a beef flavoring, and a binding agent. Each 3.5 g tablet was formulated to contain a minimum of 10 mg betulinic acid. The product was analyzed with a validated HPLC-MS method for triterpene content [6]. For the efficacy trial, the mean (SE) triterpenoid concentration was reported previously [8] as 3.22 (0.21) mg/g betulinic acid, 20 (0.01) mg/g ursolic acid, 0.24 (0.02) mg/g lupeol, 12 (0.01) mg/g β-amyrin, and 16 (0.00) mg/g α-amyrin (*n* = 3). In the placebo, the plant material was replaced with microcrystalline cellulose, as described in the safety trial [8]. Tablets prepared from the same lot were used in the stability trial (Table 1).

### 4.3. Experimental Study Design

#### 4.3.1. Animal Welfare

The animal efficacy study was placebo-controlled, randomized, and double-blinded. Procedures were designed to avoid or minimize discomfort, distress, and pain to the animals in accordance with the principles and the guidelines of the Canadian Council on Animal Care (CCAC). To ensure compliance, the experimental protocol (VRI30-12008-CE) was reviewed and approved by the test facility’s Animal Care Committee before the start of the trial.

#### 4.3.2. Blinding

The principal study investigator was blinded to experimental groups for the duration of the study. Study coordinators and laboratory personnel involved in testing and data recording were also blinded for experimental groups during the study. Lab technicians dosing the animals were the only study personnel familiar with the experimental groups, however they did not participate in data collection and analysis.

#### 4.3.3. Study Animals and Experimental Groups

The animal study was conducted at the Contract Research Organization located in Fergus, Ontario, Canada. For study purposes, the animals used in the study were laboratory, purposely bred beagle dogs. Sixty (60) beagle dogs, ranging in age from 1.4 years to 17 years and in weight from 6.1 kg to 17.8 kg, were included in the study. All dogs enrolled into the study were in good physical health, with no clinically significant health abnormalities based on prestudy examinations (physical examinations, hematology, clinical chemistry, and urinalysis). Dogs were stratified based on cortisol response to thunderstorm sounds at baseline as the primary measure, and global anxiety as the secondary measure. Dogs were considered cortisol responders if they showed 13% or greater response at the end of the session than before the thunderstorm session. Animals with elevated cortisol levels were equally randomized into experimental groups. Animals were randomized to one of six experimental groups, with ten dogs per group: (1) negative control (placebo); (2) positive control (diazepam); (3) 0.5×; (4) 1×, (5) 2×, or (6) 4× the recommended dose of test product.

#### 4.3.4. Testing Procedure

Testing procedures were performed in a room designed specifically to suit a model of noise-induced anxiety with recorded thunderstorm sounds, and to collect measures of anxiety and fear-related behaviors, such as activity or inactivity duration and frequency, and time spent in a hide box, as previously described [11]. The thunderstorm test consisted of an open field-testing room and an audio-recording of thunderstorm sounds [11]. Each testing session lasted nine minutes, where the first three minutes provided baseline data, whereas during minutes 4–6 the dogs were subjected to a taped presentation of thunder. No audible stimulus was provided during the final three minutes, which served as the post-thunder interval [11]. The entire study was designed in a manner such that each animal entered the test room on five occasions: once during the adaptation phase (no audio stimuli during 4–6 min); once during the thunder-test baseline phase (audio stimuli during 4–6 min); once during the open-field baseline phase (no audio stimuli during 4–6 min); once during the thunder-test treatment phase (audio stimuli during 4–6 min); and once during the open-field treatment phase (no audio stimuli during 4–6 min) (Table 1). The study procedures were carried out in cohorts for each phase following parallel-matched design so that each animal in each experimental group was tested with the same number of days between tests. Following thunderstorm and open-field baseline tests, dogs were dosed for five consecutive days with either placebo, diazepam, or a specific dose of the test product. After five days of dosing, all dogs were subjected to another thunderstorm test (thunder-treatment phase), followed by the open-field treatment phase, after which the anxiolytic properties of the tested products were determined.

#### 4.3.5. Assessment of Efficacy

A total of two different data categories were assessed to determine the anxiolytic properties of the test product. The first category was focused on the effect of thunder on changes in behavioral activity [11], while the second utilized serum cortisol as a physiological measure. Blood samples for cortisol analysis were taken for the two thunder tests (in the baseline and treatment phases). The other measures were taken for all four sessions (two thunderstorm and two open-field sessions; no data was collected during the adaptation phase). Blood was obtained for serum cortisol testing by venipuncture from the jugular vein at 60 min prior to the presentation of thunder and at 10 and 60 min post-thunder. Objective behavioral activity was measured by using the Ethnovision XT (Noldus, Wageningen, Netherlands) system. The following parameters served as the dependent variables for the activity analyses: activity time (total movement within the test room); inactivity time (time spent inactive); and inactivity frequency (frequency of switching from active to inactive and back).

#### 4.3.6. Statistical Analysis

Cortisol results were analyzed using repeated measures ANOVA at baseline to validate the model. Subsequently, repeated measures ANOVA that included both baseline and treatment data were carried out. For the cortisol levels and activity and observational anxiety measures, a baseline analysis was carried out to validate the model by establishing a physiological and behavioral change in response to the presentation of thunder and following the thunder interval. The data were analyzed with repeated measures ANOVA with time-interval (prethunder, thunder and post-thunder) as within-subject variables and group as between-subject variables. Tukey’s multiple comparisons were used, as appropriate, to compare different groups of data with each other. Dunnett’s test was used, as appropriate, to compare the individual groups with the control group. Baseline analysis of conditioned anxiety was assessed by comparing performance during the first three minutes of the prethunder test with performance of the first three minutes of the open-field test using a repeated measures ANOVA.

Treatment effects were first assessed by comparing presentation of thunder during the initial thunderstorm test. Furthermore, we compared the response during thunder to the response during the control open-field test. The final analysis compared the response during the baseline thunderstorm test with the response to the thunderstorm test under treatment conditions.

## Figures and Tables

**Figure 1 molecules-26-02049-f001:**
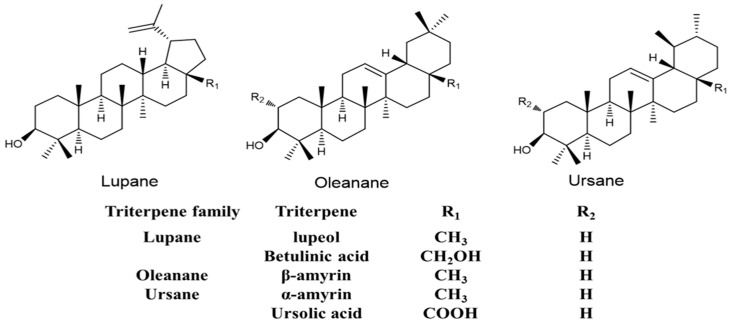
The triterpenes found in the active fraction of *Souroubea sympetala*.

**Figure 2 molecules-26-02049-f002:**
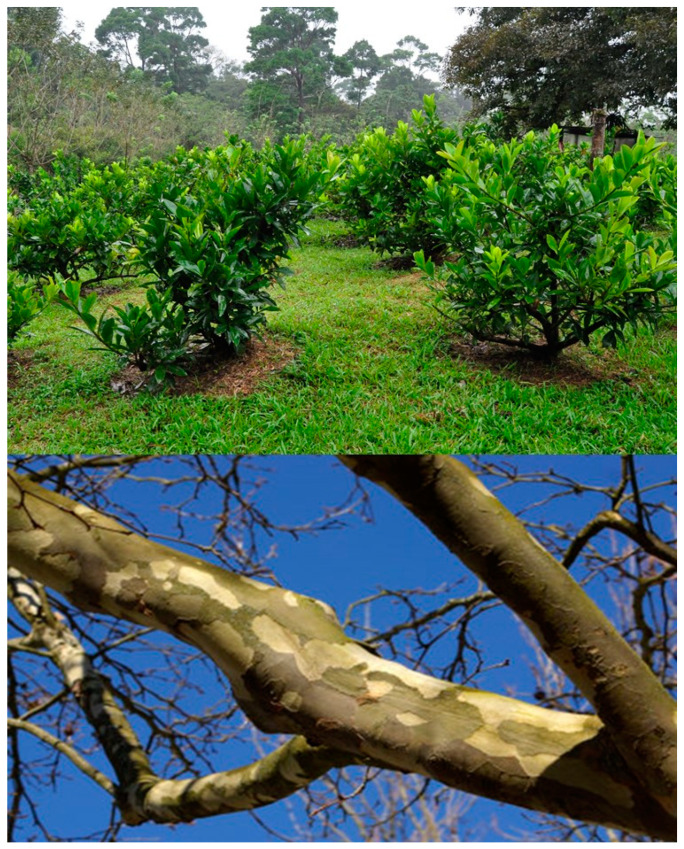
Souroubea growing sustainably in plantation (**top**) and Platanus tree trunk showing peeling bark which is harvested without harming the tree (**bottom**).

**Figure 3 molecules-26-02049-f003:**
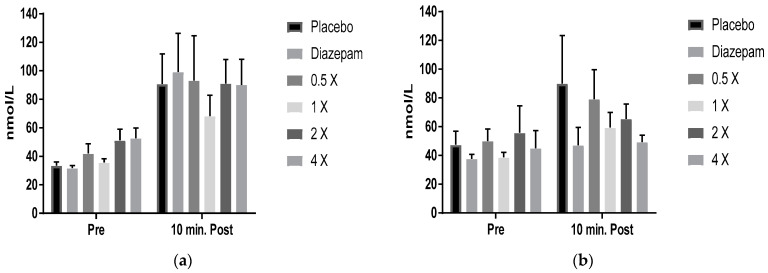
Blood cortisol concentration plotted as a function of drug treatment and temporal relationship to presentation of thunder in animals classified as responders. (**a**) Baseline data prior to drug treatments and (**b**) data during the drug treatment phase. *n* = 10 animals/group.

**Table 1 molecules-26-02049-t001:** Stability of tablets as measured by changes of active principles.

Date of Analysis	Sample	Mean (SE) Tablet Weight (g)	Mean (SE) Concentration (mg/Tablet)	
			Betulinic Acid	α-Amyrin
14 March 2016	Tablet	3.499 (0.021)	11.92 (0.472)	0.376 (0.029)
14 October 2018	Tablet	3.561 (0.012)	11.68 (0.345)	0.369 (0.024)

**Table 2 molecules-26-02049-t002:** Least square means (LSMean) of activity and inactivity time in seconds by dose, and paired difference tests of time for a specific dose versus placebo.

Dose	Inactivity LSMean	Change in Inactivity	*p*-Value	Activity LSMean	Change in Activity	*p*-Value
Placebo	114.48	-	-	24.46	-	-
0.5× *	106.48	8.01	0.6162	24.21	0.244	0.9783
1×	72.61	41.87	0.0111 ^a^	44.60	−20.14	0.0299 ^a^
2×	70.14	44.34	0.0084 ^a^	39.92	−15.47	0.0942 ^b^
4×	90.41	24.08	0.1358	45.93	−21.48	0.0199 ^a^
Diazepam	61.31	53.17	0.0016 ^a^	58.99	−34.53	0.0003 ^a^

* relative to recommended 1× dose of tablet. ^a^ Statistically significant. ^b^ Marginal significance.

## Data Availability

Data available on request.

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
