# Peer review of "Efficacy of Souroubea-Platanus Dietary Supplement Containing Triterpenes in Beagle Dogs Using a Thunderstorm Noise-Induced Model of Fear and Anxiety"

_molecules, 2021, doi:10.3390/molecules26072049_

Round 1

Reviewer 1 Report

Aleksandar Masic et al gave us a very meaningful study for botanical dietary supplement field. It is very important for pateints to improve their health using dietary supplement but not medicine. I think auhtors did a good contribution not only in food study field but also in medicine study field. This kkind of study possess a strong application promising. This is only one issue need to be clarified. Why were the dogs chose to be as research animal samples but not other animals such as rats or rabbits?

Author Response

Thank you for the encouraging comments. With respect to comment, " why did you not choose other animals such as rats or rabbits". In fact, we did  a 28  feeding trial in rats and young pigs with the botanical  materials previously and have tested extracts  in mice and fish also with no adverse effects. Based on this we felt it safe to move to a dog safety trial. Because fish and pigs are food animals, regulatory authorization would be more complex than companion animals. We have edited the text to include these comments.  In addition, this botanical bland was considered for use as a commercial product for companion animals: therefore the authors explored its efficacy in the target animal species (dogs).    

Reviewer 2 Report

An interesting design and well written paper.  Has some interest for the readers. However, before getting to that part I still have some concerns on the ethical background of this research. For example nothing is mentioned on the source of these animals that were used here. Were these lab dogs? is this still ok in Canada? Were these dogs with owners? This should be much more clearly stated, as the results could have been limited actually to the rodent models the authors are mentioning in the Introduction (first rows actually). What was the reason on going next to dogs? I don t see that discussed anywhere.

Author Response

Thank you for review and suggestions for clarifications. Here are responses to specific comments:

These were laboratory used and purposely bred beagle dogs. The study was conducted at the CRO in Canada using their colony of dogs screened for the anxiety using their noise-aversion model. The rational for using dogs over other species in efficacy study is based on intention to test botanical blend in target animal species as as a commercial product in companion animals as natural alternative to aid in anxiety.

The material and methods section is now updated to reflect use of laboratory, purposely bred beagles. 

Regarding your concerns, about safety. The botanicals are in fact traditional medicines that have been used safely by humans in other applications. In addition, we did in fact do prior safety tests with the botanicals previously in rats, for 28 under University of Ottawa animal care approval. Then later Bioniche Life Science tested the botanicals in young piglets for 28 days under European regulations and animal care approval with no adverse effects. Also fish and  mice were given extracts with no adverse effects. The previously published initial canine pilot was undertaken on three dogs (strays) showed no signs of distress and the product was then  tested in a Canadian lab colony of beagles after ethical review. Finally,  the beagles used in the present thunderstorm model, are members of a research facility for cognitive research. They are of mixed age and behavior and maintained with human interaction and facilities similar to a pet. The protocol was reviewed and no  animals were harmed during the tests. 

We have modified the text to indicate the cautious approach taken. 

Reviewer 3 Report

I found the manuscript of Masic and colleagues very interesting.

The authors have characterized in previous studies the phytoextracts deriving from Souroubea spp. Vine and evaluated the molecules with greater anxiolytic activity. Betulinic acid showed greater anxiolytic activity when combined with amyrins. For this reason, the authors have created tablets containing two phytoextracts: Souroubea spp. Vine and and Platanus spp. Bark (plant with high levels of betulinic acid). The tablets were administered to beagle dogs on which anxious behavior and cortisol levels were assessed.

The manuscript is clear and the results confirm the hypotheses made by the authors. However, some critical issues should be clarified before publication.

  • Since betulinic acid is more anxiolytic in the presence of amyrins, why did the authors not use a phytoextract with high levels of amyrins?
  • The authors should insert a graphic that explains the experimental plan and the experimental phases;
  • The figure should be improved: it would be better to insert colors to distinguish the experimental groups; authors should insert symbols between significant experimental groups;
  • The use of new formulations based on plant extracts is very important for this reason I suggest the authors to include a recent manuscript that discusses these aspects: Mastinu A et al. Protective Effects of Gynostemma pentaphyllum (var. Ginpent) against Lipopolysaccharide-Induced Inflammation and Motor Alteration in Mice. 2021;
  • Finally, it would increase the visibility of the manuscript to insert two images on the two plant species from which the extracts derive.

Author Response

Thank you for the  review and suggestions for a better manuscript which we have incorporated into the revised manuscript. 

In regards to the question of why we did not use a phytoextract with high levels of amyrins, the objective here was to use pure botanicals available to us and which had been shown to be safe in prior research. As suggested we did do a variety of mixtures to choose the best ratio ( now discussed more clearly in the revised version)  The recommendation of the high amyrin extract is a good suggestion for future research which was included in the discussion.A discussion has been added citing Mastinu A et al. as recommended to show that a variety of studies on botanical alternatives for the animal health market are being developed .

The suggestion of colour images was made. We have added Souroubea Platanus  images, as suggested. Our preference for  figures is greyscale. 

Round 2

Reviewer 2 Report

yes. The authors did not convince me. Reject from me.